# The Effect of Statins on Ocular Disorders: A Systematic Review of Randomized Controlled Trials

**DOI:** 10.3390/ph16050711

**Published:** 2023-05-07

**Authors:** Charoula Lymperopoulou, Stylianos A. Kandarakis, Ismini Tzanaki, Ioanna Mylona, Theodoros Xanthos, Aris P. Agouridis

**Affiliations:** 1School of Medicine, European University Cyprus, Nicosia 2404, Cyprus; c.limperopoulou@gmail.com (C.L.); isminitzan@yahoo.gr (I.T.); 2Department of ophthalmology, National and Kapodistrian University of Athens, 1st University Eye Clinic, G. Gennimatas General Hospital, 11527 Athens, Greece; s.kandarakis@gmail.com; 3Department of Ophthalmology, General Hospital of Serres, 62210 Serres, Greece; milona_ioanna@windowslive.com; 4School of Health Sciences, University of West Attica, 10434 Athens, Greece; txanthos@uniwa.gr; 5Department of Internal Medicine, German Oncology Center, Limassol 4108, Cyprus

**Keywords:** statins, cataract, age-related macular disease (AMD), glaucoma, diabetic retinopathy, uveitis, dry eye disease

## Abstract

Aim: Statins have been established in the market not only due to their ability to lower plasma cholesterol levels but also due to their pleiotropic effects. In the literature, there is a controversy regarding the role of statins in ophthalmology. We aimed to systematically address the possible effect of statin therapy on ocular diseases and to identify if there is a beneficial relationship. Methods: We searched PubMed and Cochrane Library databases up to 31 December 2022 for studies evaluating the effect of statins on ocular diseases. We included all relevant Randomized Control Trials (RCTs) that have been conducted in the adult population. PROSPERO registration number: CRD42022364328. Results: Nineteen RCTs were finally considered eligible for this systematic review, with a total of 28,940 participants. Ten studies investigated the role of simvastatin, suggesting a lack of cataractogenic effect and a possible protective role in cataract formation, retinal vascular diseases, and especially diabetic retinopathy, age-related macular disease progression, and non-infectious uveitis. Four studies investigated lovastatin, showing no cataractogenic effect. Three studies examined atorvastatin, revealing conflicting results regarding diabetic retinopathy. Two studies examined rosuvastatin, indicating a possibly harmful effect on lenses and a significant protective effect on retinal microvasculature. Conclusions: Based on our findings, we believe that statins have no cataractogenic effect. There are indications that statins may have a protective role against cataract formation, AMD, diabetic retinopathy progression, and non-infectious uveitis. However, our results were insufficient for any robust conclusion. Future RCTs, with large sample sizes, on the current topic are therefore recommended to provide more solid evidence.

## 1. Introduction

Nowadays, multiple statins are available in the market, including atorvastatin, rosuvastatin, simvastatin, lovastatin, pravastatin, fluvastatin, and pitavastatin [1,2]. Since 1987, when the first statin was introduced, intensive clinical investigations have proven that statin therapy is well tolerated, with an excellent safety profile, although some adverse events have been reported, with the most common being muscle toxicity and elevation of liver enzymes [3,4]. The occurrence of other adverse effects, such as the slightly elevated risk of newly diagnosed diabetes mellitus and the possible increased risk of hemorrhagic stroke, has limited importance compared to the proven cardiovascular benefits of statin therapy [3,4].

Many publications during the past have focused mainly on statin’s therapeutic effect in cholesterol-lowering. However, several recent clinical trials suggest that statin therapy benefits are partially associated with other cholesterol-lowering independent effects, known as pleiotropic effects [5]. Statin’s most known pleiotropic effects include anti-inflammatory and immunomodulatory properties, antioxidant activity, neuroprotective actions, the ability to reduce thrombogenicity, and the improvement of endometrial dysfunction [5,6].

The role of statin therapy in ophthalmology has not been clarified yet. Due to their pleiotropic effects, relatively safe profile, and low cost, statins seem to be an up-and-coming option for preventing and managing ocular diseases [7]. However, studies that examined the association between statin therapy and eye disorders have provided conflicting results [8]. 

This study aims to perform a systematic review of RCTs, assessing the association of ocular conditions, including cataracts, age-related macular disease (AMD), glaucoma, diabetic retinopathy, dry eye, and uveitis, with statin therapy.

## 2. Results

### 2.1. Study Selection 

Figure 1 summarizes the results of our extended literature search in a Preferred Reporting Items for Systematic Reviews and Meta-Analyses (PRISMA) chart [9]. We identified 567 publications through PubMed, Cochrane Library, and the relevant studies’ reference lists. Of these, 303 publications were duplicates in the database, and an additional 239 records were excluded after reviewing their titles and abstracts. The full texts of the remaining 26 studies were further assessed for eligibility. Studies to be included in this review had to match predetermined criteria according to the Population, Intervention, Comparison, Outcomes, and Study (PICOS) approach. Criteria for inclusion and exclusion are specified in Appendix A. After reviewing the full texts, seven more studies were excluded for not meeting the inclusion criteria (Appendix A) [10,11,12,13,14,15,16]. After the final exclusion, 19 RCTs [17,18,19,20,21,22,23,24,25,26,27,28,29,30,31,32,33,34,35] fulfilled the criteria to be included in our systematic review. 

### 2.2. Study Characteristics

The studies that met the eligibility criteria for inclusion in our systematic review examined a total of 28,940 participants, who were followed up for a mean of 24.4 months. The mean age of all participants was approximately 55 years old, and the male-to-female ratio was similar. We initially searched for seven different statins, but in the included studies, only four were evaluated: simvastatin (10 studies), atorvastatin (3 studies), lovastatin (4 studies), and rosuvastatin (2 studies). Among the studies, ten analyzed the effect of statins in the human lens, and seven studies investigated the role of statins in the progression of retinal vascular diseases focusing mainly on diabetic retinopathy. The remaining studies examined other eye disorders, such as AMD and non-infectious uveitis. The studied population mainly suffered from hyperlipidemia and diabetes mellitus. Other comorbidities that were also reported were non-advanced AMD and asymptomatic aortic stenosis, and a percentage of the participants was considered to be at high risk for coronary heart diseases. A series of ophthalmological tests (slit lamp examination, measurement of visual acuity, fundoscopy) and a biochemical investigation were evaluated in most of the trials. More specific tests were also performed based on the aim of each study. No significant adverse events have been reported except for some minor symptoms that can be attributed as possible side effects of statins, such as muscle pain, weakness, rash, and mild headache. The studies were conducted in different areas around the world, including the US (5 studies), India (3 studies), Australia (3 studies), Europe (4 studies), Turkey (1 study), Russia (1 study), and China (1 study). Table 1 lists the characteristics of each one of the included studies. 

### 2.3. Outcomes of the Included Studies 

The results are summarized in Table 1. According to our analysis, 10 studies investigated the role of simvastatin, suggesting a lack of cataractogenic effect and a possible protective role in cataract formation, retinal vascular diseases, and especially diabetic retinopathy, age-related macular disease progression, and non-infectious uveitis. In addition, four studies investigated lovastatin, showing no cataractogenic effect. Furthermore, three studies examined atorvastatin, revealing conflicting results regarding diabetic retinopathy, while two studies examined rosuvastatin, indicating a possibly harmful effect in lenses and a significant protective effect on retinal microvasculature.

### 2.4. Quality Appraisal

The bias risk was assessed using version 2 of the Cochrane risk of bias tool for randomized trials (RoB2) [36] (Figure 2 and Figure 3). Out of the 19 studies assessed, only two studies were judged to have “high” bias risk, while six were judged to have “some concerns”. All the 11 remaining trials were judged to have a “low” bias risk.

## 3. Discussion

To the best of our knowledge, this is the first systematic review conducted on RCTs that examined the relation between statin use and ocular diseases in the adult population. Our findings suggested conflicting results regarding the effect of statins on the eyes.

Currently, the treatment options for cataracts, AMD, glaucoma, diabetic retinopathy, dry eye, and uveitis are limited, despite affecting a large percentage of the population worldwide. Further options of medical therapy are required to slow the damage in vision loss and, at the same time, to prevent the disease from occurring [7,8]. Due to their pleiotropic effects, relatively safe profile, and low cost, statins seem to be an up-and-coming option for preventing and managing ocular diseases. Statins have already been recommended to use in chronic inflammatory disorders such as rheumatoid arthritis and systemic lupus erythematosus [5]. However, in the literature, the studies that examined the association between statin therapy and eye disorders have provided controversial results [8]. 

The exact mechanism linking statin and ocular diseases has not been clarified yet. Their ability to reduce pro-inflammatory cytokines may minimize the harm of inflammation in tissue damage in wet AMD, diabetic retinopathy, dry eye disease, and uveitis [7]. One of the RCTs that have addressed the anti-inflammatory properties of statins was the PRINCE (pravastatin inflammation/CRP evaluation) trial, which noted that statins were involved in the reduction of C-reactive protein (CRP) in patients suffering from cardiovascular diseases [37]. Moreover, in the JUPITER (Justification for the Use of Statins in Prevention: an Intervention Trial Evaluating Rosuvastatin) study, it was first reported that rosuvastatin administration reduced cardiovascular morbidity and mortality, as well as overall mortality, in apparently healthy subjects without hyperlipidemia, but with elevated high-sensitivity CRP (hs-CRP) levels [38].

Statins also have the ability to reduce transforming growth factor beta (TGF-β) and Rho-kinase inhibitory activity, which may have a beneficial impact on glaucoma by improving the aqueous outflow [7,39]. Furthermore, statins decrease vascular endothelial growth factor (VEGF) expression and nicotinamide adenine dinucleotide phosphate oxidase (NADPH) oxidase inhibition, which allows a significant increase in endothelial structure and function, that may benefit the progression of diabetic retinopathy and AMD [7]. Of note, oxidative stress is a significant risk factor for age-related cataracts, further enhancing a possible connection with statins due to their known antioxidative effects [39,40].

Most trials investigated the reaction of the human lens in statin therapy, although a clear causal association could not be ascertained. A meta-analysis that used data from fourteen clinical trials (eight observational studies and six randomized), including approximately 2,403,644 patients and 25,618 cataracts, showed that statin use was related to a 19% drop in the risk of cataracts (OR 0.81, 95% CI: 0.72–0.92, *p* = 0.0009) [41]. The effect appeared to be statistically significant for clinical cataracts (19% decrease, OR 0.81, 95% CI 0.71–0.93, *p* = 0.0022), but it was not statistically significant for the lenticular opacities (OR 0.81, 95% CI 0.59–1.12, *p* = 0.2106) [41]. Of note, the effect observed in the RCTs had the same importance as the observational studies, but it was not statistically significant (OR 0.84, 95% CI 0.67–1.05, *p* = 0.1189) [41]. Overall, a clinically relevant and statistically significant protective role of statins was demonstrated by the analysis of the results, which appeared to be more pronounced in younger patients and with a more extended follow-up period [41]. A more recent meta-analysis, conducted by Yu et al., analyzed the results of seventeen studies, including six cohorts, six case controls, and five RCTs, in more than 313,200 patients, aiming to reach a clear conclusion to the debatable cataractogenic effect of statins [42]. Analysis of the included cohort studies showed that the pooled RR was 1.13 (95% CI, 1.01–1.25), which suggested that statins were responsible for a 13% increased risk of cataract formation or cataract surgery [42]. In contrast, the pooled RRs of case-control studies and RCTs were 1.10 (95% CI, 0.99–1.23) and 0.89 (95% CI, 0.72–1.10), respectively, which indicated that no link was detected between statin use and the risk of cataract development or surgery for cataract extraction [42]. Significant heterogeneity was detected among both cohort studies and case-control studies, although RCTs had low heterogeneity scores [42]. The researchers concluded that there was insufficient data to establish a cataractogenic effect of statins [42].

Concerning the role of statins in the progression of retinal vascular diseases, our results supported that statins may have the ability to minimize diabetic retinal complications either by interfering with retinal vessels or by reducing the hard exudates [26,27,28,31,35]. Miyahara et al. reported that high-dose simvastatin could decrease the expression of vascular endothelial growth factor (VEGF), which has an essential role in the pathogenesis of both diabetic retinopathy and exudative AMD [43]. Additionally, Weis et al. noticed that endothelial release of VEGF was considerably reduced with high statin concentrations, but no significant difference was detected with low dosages of statins [44]. 

This protective effect of statins in diabetic retinopathy is further enhanced by a recent meta-analysis that used data from six cohort studies with a total of 558,177 patients [45]. Analysis showed that statin therapy was correlated with a reduced prevalence of diabetic retinopathy [HR: 0.68 (0.55, 0.84), *p* < 0.001; I^2^: 95%] [45]. Regarding the subtypes of diabetic retinopathy, statins reduced the risk of proliferative diabetic retinopathy [HR:0.69 (0.51, 0.93), *p* = 0.01; I^2^: 90%], non-proliferative diabetic retinopathy [HR: 0.80 (0.66, 0.96), *p* = 0.02; I^2^: 93%], and diabetic macular edema [HR: 0.56 (0.39, 0.80), *p* = 0.002; I^2^: 82%] [45]. Furthermore, the findings suggested that statin use minimizes the necessity of more interventional techniques [HR:0.72 (0.64, 0.80), *p* < 0.001; I^2^: 73%], such as retinal laser treatment, intravitreal injection with anti-VEGF, and vitrectomy [45].

Regarding the association between statins and AMD risk, Guymer et al. addressed that simvastatin may retard the progression of non-advanced AMD [30]. In support of this notion, a meta-analysis using 15 articles (seven cohort studies, five case-control studies, and three cross-sectional studies), with the number of subjects ranging from 744 to 104,176, indicated no important connection between statin use and the risk of any AMD (RR, 0.95; 95% CI, 0.74–1.15) [46]. Seven studies assessed data on the relationship between statins and early AMD, with a total of 27,308 participants [46]. When the results of these studies were analyzed, the authors found that statins can drop the incidence of early AMD by approximately 17% (RR, 0.83; 95% CI, 0.66–0.99) [46]. The remaining eight studies, with a total of 22,973 participants, reported the role of statins in late AMD [46]. The analysis of the results showed an important protective effect of statins on exudative AMD (RR, 0.90; 95% CI, 0.80–0.99), although no link was detected between statins and geographic atrophy (RR, 1.16; 95% CI, 0.77–1.56) [46]. In general, this meta-analysis supported that statins had a protective role for both early and exudative AMD [46]. However, a recent systematic review and meta-analysis that aimed to link statins and AMD led to a different conclusion [47]. Researchers reviewed a total of 22 studies, with 2,063,195 participants, 15.2% of whom were diagnosed with AMD. The overall OR of AMD in statin-receiving participants was 0.93 (95% CI; 0.83–1.05, *p* = 0.225). The OR of AMD in those that received statins were 0.92 (95% CI; 0.75–1.13, *p* = 0.440) in case-control studies, 0.95 (95% CI; 0.82–1.09, *p* = 0.458) in cohort studies, and 0.951 (95% CI; 0.59–1.53, *p* = 0.831) in cross-sectional studies. This meta-analysis showed that statin therapy has no positive or negative impact on AMD development [47]. 

Patients with uveitis usually have to follow an intensive steroid treatment plan for a prolonged time period to avoid any relapse of the disease [39]. Shrinsky et al.’s findings supported the hypothesis that statins have the ability to reduce the extent of ocular inflammation in uveitis due to their anti-inflammatory and immunomodulatory properties [34]. Additionally, a retrospective population-based case-control study selected 108 incident cases of uveitis, with most of them being anterior (81%) and non-infectious (76%) [48]. Comparing the participants with non-infectious uveitis and their respective general population controls, the percentage of those suffering from uveitis was almost 56% less in statin users in contrast to non-statin users (OR: 0.44, 95% CI: 0.22 to 0.88, *p* = 0.02) when adjusting for multiple factors including age, gender, race, smoking status, and autoimmune diseases [48].

Of note, no RCT has examined the association between statins and glaucoma. However, in a recent systematic review and meta-analysis [49] that included 17 studies, mainly cohort and case-control studies, with a total of 515,788 patients, statin use was associated with a slightly lower risk of open-angle glaucoma onset, while no association between statin use and open-angle glaucoma progression was observed. The use of underpowered studies, however, weakened the overall meta-analysis outcome [49].

### Limitations

This systematic review examined the association between statin therapy and ocular diseases. We believe that through an extensive search of two databases, we were able to detect and analyze all the relevant RCTs. However, certain limitations should be acknowledged. Different classifications for the evaluation of the findings were used in the studies, making it hard to compare them. Additionally, due to high heterogeneity, we were unable to perform a meta-analysis.

## 4. Materials and Methods

### 4.1. Study Design

We performed a qualitative synthesis of published randomized controlled trials (RCTs) to address the possible effect of statin therapy, including simvastatin, lovastatin, atorvastatin, fluvastatin, pravastatin, rosuvastatin, and pitavastatin on ocular diseases, namely, cataracts, AMD, glaucoma, diabetic retinopathy, dry eye, and uveitis, and to identify if there is any beneficial relationship.

### 4.2. Search Strategy

This systematic review was registered on PROSPERO (ID number: CRD42022364328) and was conducted following PRISMA recommendations [9]. An extensive bibliographic search of PubMed and Cochrane library databases was conducted until 31 December 2022, using the following keywords: ‘‘simvastatin and cataract”, ‘‘simvastatin and lens opacities”, “simvastatin and glaucoma”, “simvastatin and diabetic retinopathy”, “simvastatin and dry eye”, “simvastatin and uveitis”, “simvastatin and AMD”, “simvastatin and macular edema”, “lovastatin and cataract”, ‘‘lovastatin and lens opacities”, “lovastatin and glaucoma”, “lovastatin and diabetic retinopathy”, “lovastatin and dry eye”, “lovastatin and uveitis”, “lovastatin and AMD”, “lovastatin and macular edema”, “atorvastatin and cataract”, ‘‘atorvastatin and lens opacities”, “atorvastatin and glaucoma”, “atorvastatin and diabetic retinopathy”, “atorvastatin and dry eye”, “atorvastatin and uveitis”, “atorvastatin and AMD”, “atorvastatin and macular edema”, “fluvastatin and cataract”, ‘‘fluvastatin and lens opacities”, “fluvastatin and glaucoma”, “fluvastatin and diabetic retinopathy”, “fluvastatin and dry eye”, ”fluvastatin and uveitis”, “fluvastatin and AMD”, “fluvastatin and macular edema”, “pravastatin and cataract”, ‘‘pravastatin and lens opacities”, “pravastatin and glaucoma”, “pravastatin and diabetic retinopathy”, “pravastatin and dry eye”, “pravastatin and uveitis”, “pravastatin and AMD”, “pravastatin and macular edema”, “rosuvastatin and cataract”, ‘‘rosuvastatin and lens opacities”, “rosuvastatin and glaucoma”, “rosuvastatin and diabetic retinopathy”, “rosuvastatin and dry eye”, “rosuvastatin and uveitis”, “rosuvastatin and AMD”, “rosuvastatin and macular edema”, “pitavastatin and cataract”, ‘‘pitavastatin and lens opacities”, “pitavastatin and glaucoma”, “pitavastatin and diabetic retinopathy”, “pitavastatin and dry eye”, “pitavastatin and uveitis”, “pitavastatin and AMD”, “pitavastatin and macular edema”. Additionally, the reference lists of eligible studies were manually searched to detect any relevant articles that could meet the PRISMA criteria. This search has included only papers written in English. 

### 4.3. Eligibility Criteria 

Eligible studies for our systematic review were RCTs in adults (>18 years old) that compared statin therapy (simvastatin, lovastatin, atorvastatin, fluvastatin, pravastatin, rosuvastatin, and pitavastatin) versus placebo, diet, or control group. We excluded any nonrandomized studies, including case controls, cohort, and cross-sectional studies, articles written in a non-English language, articles focusing on the pediatric population (<18 years old), articles with insufficient data, and articles irrelevant to our primary aim. 

### 4.4. Data Extraction 

Two authors (C.L., A.P.A.) independently scanned the abstract, title, or both, of every record and assessed the full text to determine which studies meet the inclusion criteria in order to be included in the review. During the process, any disagreement between the reviewers was resolved by consensus discussion. Data extraction was done in accordance with the PRISMA model. Because of the study’s design, there was no need either for approval by the National Bioethics Committee (CNBC) or for informed permission from the patients. Initially, all the search results to databases were screened for duplications through Zotero software. After removing duplicates, data were extracted using Excel and included the following information: first author, publication year, the country where the trial was conducted, number and characteristics of the participating population, study duration, statin therapy, comparator, and the outcome. Following that, papers were chosen based on a further review of full-text articles to conclude our final selection. 

### 4.5. Assessment of Risk of Bias 

A Risk of Bias Assessment was performed for each included study to establish transparency of systematic review results and findings, using the RoB2 tool for randomized trials [36]. The assessment is divided into a series of domains through which bias might be introduced into the trial. Within each domain, a number of questions (‘signaling’ questions) aim to facilitate judgments regarding the risk of bias. Based on the answers to the signaling questions, an algorithm generates a proposed judgment regarding the risk of bias from each domain. Judgment can be expressed as having a “Low” risk of bias, “High” risk of bias, or “Some concerns”. 

## 5. Conclusions

This systematic review found 19 RCTs that assess the effect of statin use (simvastatin, lovastatin, atorvastatin, and rosuvastatin) in eye disorders (cataracts, AMD, retinal vascular disease, and non-infectious uveitis). Based on our findings, we believe that statins have no cataractogenic effect. There are indications that statins may have a protective role against cataract formation, AMD, diabetic retinopathy progression, and non-infectious uveitis. However, our results were insufficient for any robust conclusion. Future research is required to give definite answers regarding the role of statins in ophthalmology.

## Figures and Tables

**Figure 1 pharmaceuticals-16-00711-f001:**
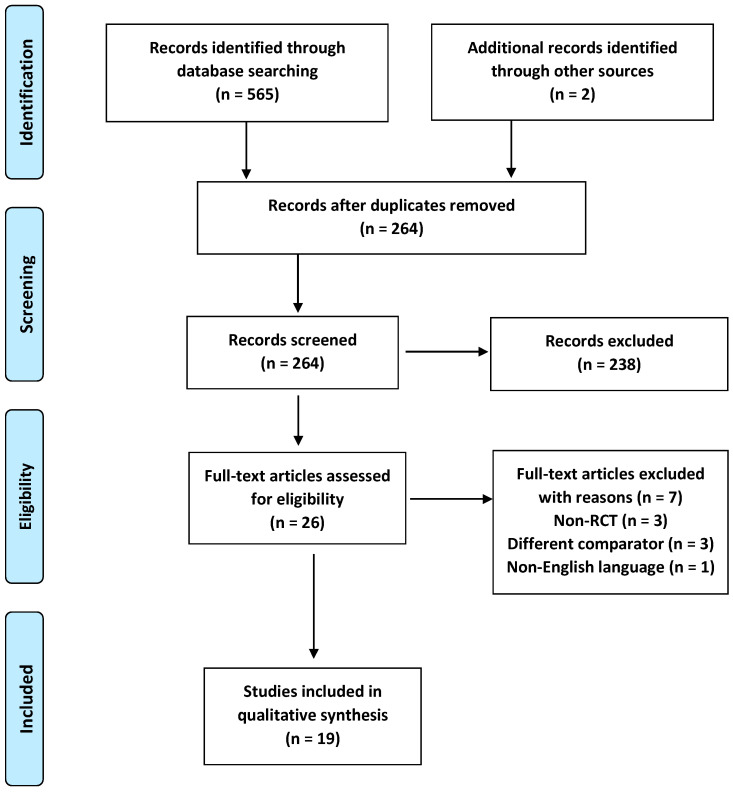
The PRISMA flowchart of our systematic review.

**Figure 2 pharmaceuticals-16-00711-f002:**
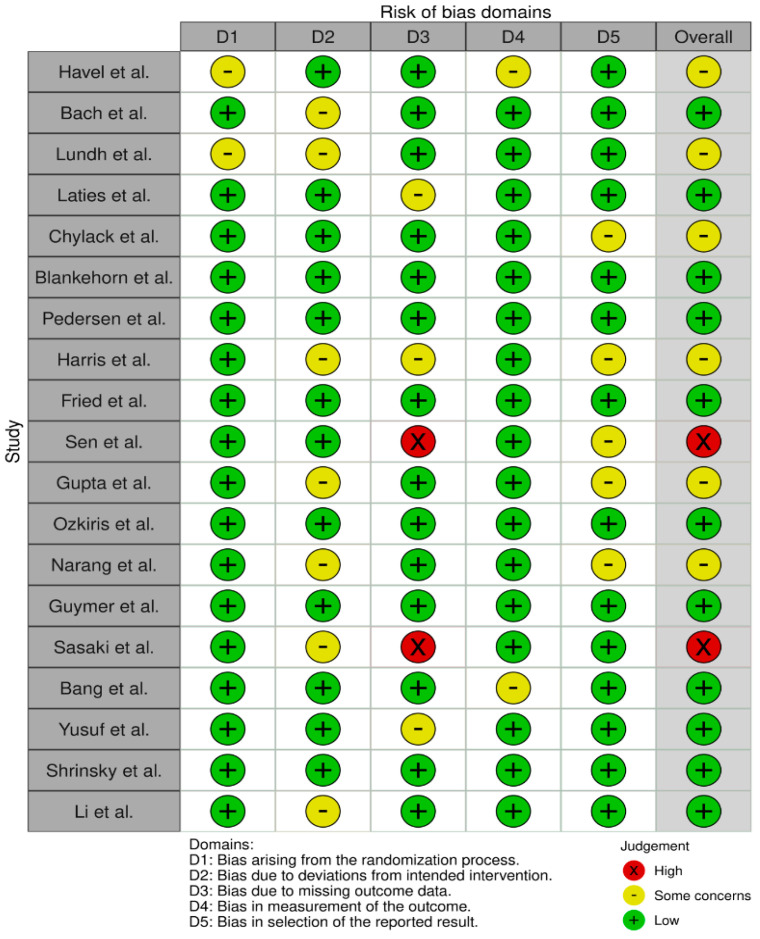
Traffic light plot for risk of bias assessment of the included studies using the revised Cochrane risk of bias tool for randomized trials (RoB-2) [17,18,19,20,21,22,23,24,25,26,27,28,29,30,31,32,33,34,35].

**Figure 3 pharmaceuticals-16-00711-f003:**
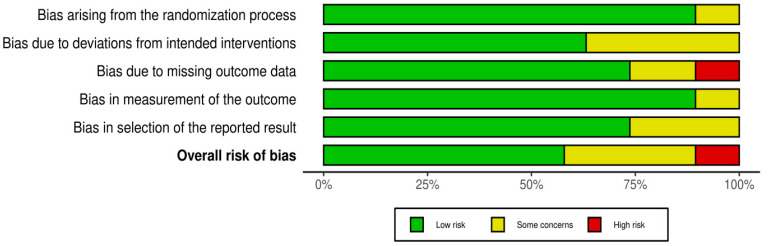
Summary plot for risk of bias assessment of the included studies using the revised Cochrane risk of bias tool for randomized trials (RoB-2).

**Table 1 pharmaceuticals-16-00711-t001:** Characteristics of the included studies.

First Author	Year	Country	Participants (No)	Study Duration	Comorbidities	Mean Age (Years)	Ocular disorder	Statin Therapy (Dosage)	Comparator	Outcomes	*p*-Value
Havel [17]	1987	US	101	6 weeks	Heterozygous familial hypercholesterolemia	44	Cataract	Lovastatin (5–40 mg twice/day or 20 to 40 once/day)	Placebo	No change in the prevalence of lens opacities	N/A
Bach [18]	1990	Australia	20	4 weeks	Hyperlipidemia, coronary artery disease	N/A	Cataract	Simvastatin (2.5, 5, 10, or 20 mg/day)	Placebo	No difference regarding lens opacities and visual acuity	N/A
Lundh [19]	1990	Sweden	29	2 years	Hypercholesterolemia	N/A	Cataract	Simvastatin (10–20 mg twice/day)	Control	No harmful effect in the human lens	*p* = N/S
Laties [20]	1991	US	8245	48 weeks	Hypercholesterolemia	55	Cataract	Lovastatin (20 or 40 mg once or twice/day)	Placebo	No effect on the human lens	*p* = N/S
Chylack [21]	1993	US	192	2 years	Hypercholesterolemia	53.5	Cataract	Lovastatin (40 mg/day)	Placebo	Cataract progression showed no significant difference between the two groups	*p* = N/S (Only for nuclear cataract in right eyes, *p* < 0.02)
Blankenhorn [22]	1993	US	270	4 years	Coronary artery disease	58	Cataract	Lovastatin (40 mg twice/day)	Placebo	No difference was found between groups in new onset or worsening of lens opacities	N/A
Pedersen [23]	1994	Scandinavian countries	4444	5.4 years	Angina pectoralis or previous myocardial infarction, hypercholesterolemia	58.9	Cataract	Simvastatin (20–40 mg once/day)	Placebo	No difference between the two groups	*p* = 0.19
Harris [24]	1995	UK	474	18 months	High risk of coronary heart disease	53.9	Cataract	Simvastatin (20 or 40 mg/day)	Placebo	No differences between the two groups	N/A
Fried [25]	2000	US	39	2 years	Insulin-dependent diabetes mellitus	32.1	Diabetic Retinopathy	Simvastatin	Diet	No difference in change in retinopathy status	N/A
Sen [26]	2001	India	50	180 days	Diabetes mellitus, hypercholesterolemia	53.9	Diabetic Retinopathy	Simvastatin (20 mg/day)	Placebo	Delay the progression of diabetic retinopathy	*p* = 0.009
Gupta [27]	2004	India	30	18 weeks	Diabetes, Hyperlipidemia, macular edema with hard exudates	54.1	Diabetic macular edema	Atorvastatin (10 mg/day)	Control	Reduction of hard exudates and none of the patients suffered from subfoveal lipid migration	*p* = 0.07
Ozkiris [28]	2006	Turkey	45	10 weeks	Diabetes type 2	58.3	Diabetic Retinopathy	Atorvastatin (10 mg/day)	Placebo	Vascular resistance improvement, decrease of mean peak systolic velocity of the ophthalmic artery and the central retinal artery	*p* < 0.05
Narang [29]	2012	India	30	6 months	Non-insulin-dependent diabetes, non-proliferative diabetic retinopathy with clinically significant macular edema (CSME)	55.9	CSME	Atorvastatin (20 mg/day)	Placebo	No significant difference in visual acuity, macular edema, and hard exudate among the two groups	*p* = 0.39 for visual acuity *p* = 0.62 for macular edema
Guymer [30]	2013	Australia	114	6 years	Non-advanced AMD	74.6	AMD	Simvastatin (40 mg/day)	Placebo	Simvastatin retarded the AMD progression compared to the placebo group	*p* = 0.047
Sasaki [31]	2013	Australia	102	3 years	Non-advanced AMD	N/A	Retinal vascular diseases	Simvastatin (40 mg/day)	Placebo	Simvastatin group had a significantly larger retinal arteriolar caliber compared to the control group	*p* = 0.443
Bang [32]	2015	Denmark	1873	4.3 years	Asymptomatic aortic stenosis	67.5	Cataract	Simvastatin (40 mg/day)	Placebo	Simvastatin plus ezetimibe was associated with a 44% lower risk of cataract	*p* = 0.034
Yusuf [33]	2016	N/A	12,705	5.6 years	Cardiovascular risk factors	65.7	Cataract	Rosuvastatin (10 mg/day)	Placebo	More patients in the rosuvastatin group needed cataract surgery compared to the placebo group	*p* = 0.02
Shrinsky [34]	2017	Russia	50	2 months	Active non-infectious uveitis	43.9	Non-infectious uveitis	Simvastatin (40 mg/day)	Control	Patients in the simvastatin group received significantly less steroid treatment and showed an improvement in visual acuity and reduction in ocular inflammation.	*p* < 0.001
Li [35]	2019	China	127	12 months	Hypercholesterolemia	53.7	Retinal vascular diseases	Rosuvastatin (10 mg/day)	Control	A significant effect of rosuvastatin on retinal microvasculature was detected, including artery vein ratio increase, venular constriction, and arteriolar dilation	*p* < 0.01

N/A: Not available.

## Data Availability

Data sharing not applicable.

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
