# Peer review of "The Effect of Statins on Ocular Disorders: A Systematic Review of Randomized Controlled Trials"

_pharmaceuticals, 2023, doi:10.3390/ph16050711_

Round 1

Reviewer 1 Report

Lymperopoulou et al. described the effect of statins on ocular disorders. Authors aimed to systematically address the possible effect of statin therapy on ocular diseases and to identify if there is a beneficial relationship. Based on their findings, they are uncertain about the effect of statins on eye disorders. Although the topic is incredibly intriguing, there are certain issues that need to be addressed.

1)     Since the authors declare that they are uncertain about the effect of statins on eye disorders, the title should be changed to The Negative Effect of Statins on Ocular Disorders: A Systematic Review of Randomized Controlled Trials

2)     Figure 3 resolution should be improved

3)     Discussion part need more interpretation  

4)     The manuscript contains several grammar mistakes and should be revised carefully.

5)     Study limitations should be addressed in a separate subtitle.

6)     3. Eligibility criteria section 4.3 should be illustrated in a separate figure.

  1. The manuscript contains several grammar mistakes and should be revised carefully.

Reviewer 2 Report

The paper „The Effect of Statins on Ocular Disorders: A Systematic Review
of Randomized Controlled Trials“ by Charoula Lymperopoulou et al. provides a review about statin therapy and a possible effect of ocular disorders.

The authors performed a systematic literature research of randomized controlled trials and statin therapy with simvastatin, lovastatin, atorvastatin, fluvastatin, pravastatin, rosuvastatin or pitavastatin and ocular diseases including cataract, AMD, glaucoma, diabetic retinopathy, dry eye and uveitis. They provided a thorough overview of their search strategy, eligibility criteria and data extraction. Results were presented by describing their characteristics and outcomes as summarized in table 1. They found that there were some indications that statins may have a protective role against cataract formation, AMD, diabetic retinopathy progression and non-infectious uveitis, but the results were insufficient for any conclusion.

Since this is a review, it would be interesting to understand, why the authors chose to relate statin-therapy to ocular diseases.

What makes statins interesting to ophthalmology in general and why did the authors choose specifically the mentioned ocular diseases? Did they suspect a specific metabolic mechanism related to statins?

The methods section should be placed before the results to improve readability.

Reviewer 3 Report

1.     Abstract; rewrite in terms of discussion.

2.     Line 17 superscript ..31st of December 2022

3.     In the discussion, authors mentioned that ’’Miyahara et al. reported that high-dose simvastatin can decrease the expression of vascular endothelial growth factor (VEGF), which has an essential role in the pathogenesis of both diabetic retinopathy and exudative AMD [39]. Additionally, Weis et al. noticed that endothelial release of VEGF was considerably reduced with high statins concentrations, but no significant difference was detected with a low dosages of statins [40].’’ But in the Abstract and conclusion authors mentioned that ‘’we are uncertain about the effect of statins on eye disorders.’’ These indicate contradictory conclusions about statins in eye disorders, Justify?

4.     Add statistical analysis of different statins data in terms of treatment and betterment of diseases.

5.     Add recent data and analysis.

6.     Rewrite the conclusion as per obtained results and conclusions should not be ambiguous with discussion.

Moderate editing of English language.
